# ROLE-WISE DATA AUGMENTATION FOR KNOWLEDGE DISTILLATION

## ABSTRACT

Knowledge Distillation (KD) is a common method for transferring the "knowledge" learned by one machine learning model (the *teacher*) into another model (the *student*), where typically, the teacher has a greater capacity (e.g., more parameters or higher bit-widths). To our knowledge, existing methods overlook the fact that although the student absorbs extra knowledge from the teacher, both models share the same input data – and this data is the only medium by which the teacher's knowledge can be demonstrated. Due to the difference in model capacities, the student may not benefit fully from the same data points on which the teacher is trained. On the other hand, a human teacher may demonstrate a piece of knowledge with individualized examples adapted to a particular student, for instance, in terms of her cultural background and interests. Inspired by this behavior, we design data augmentation agents with distinct roles to facilitate knowledge distillation. Our data augmentation agents generate distinct training data for the teacher and student, respectively. We focus specifically on KD when the teacher network has greater precision (bit-width) than the student network.

We find empirically that specially tailored data points enable the teacher's knowledge to be demonstrated more effectively to the student. We compare our approach with existing KD methods on training popular neural architectures and demonstrate that role-wise data augmentation improves the effectiveness of KD over strong prior approaches. The code for reproducing our results will be made publicly available.

## 1 INTRODUCTION

**Background and Motivation.** In the educational psychology literature, it is generally considered beneficial if teachers can adapt curricula based upon students' prior experiences (Bandura, 2002; Brumfiel, 2005; Gurlitt et al., 2006; Slotta & Chi, 2006). These vary widely depending on students' cultural backgrounds, previous educational experiences, interests, and motivations. To help students with different prior experiences to comprehend, memorise, and consolidate a piece of knowledge, teachers may provide extra and customized teaching material during their teaching processes. For instance, when teaching the concept of the color *pink*, a teacher may choose flamingos, sakura (cherry blossoms), or ice cream cones as the example, depending on a student's background.

Knowledge distillation (KD) (Bucilua et al., 2006; Hinton et al., 2014) is a common framework for training machine learning models. It works by transferring knowledge from a higher-capacity teacher model to a lower-capacity student model. Most KD methods can be categorized by how they define the knowledge stored in the teacher (i.e., the "soft targets" of training as defined in existing literature). For instance, Hinton et al. (2014) originally proposed KD for neural networks, and they define the output class probabilities (i.e., soft labels) generated by the teacher as the targets for assisting the training of students. In a follow up work, Romero et al. (2015) defined the soft targets via the feature maps in the teacher model's hidden layers.

To train a student network with KD effectively, it is important to distill as much knowledge from the teacher as possible. However, previous methods overlook the importance of the *medium* by which the teacher's knowledge is demonstrated: the training data points. We conjecture that there exist examples, not necessarily seen and ingested by the teacher, that might make it easier for the student to absorb the teacher's knowledge. Blindly adding more training examples may not be beneficial

because it may slow down training and introduce unnecessary biases (Ho et al., 2019). The analogy with how human teachers adjust their teaching to their students' particular situations (e.g., with the feedback gathered from the students during teaching) suggests that a reasonable yet uninvestigated approach might be to augment the training data for both the teacher and student according to *distinct* policies.

In this paper, we study whether and how adaptive data augmentation and knowledge distillation can be leveraged synergistically to better train student networks. We propose a two-stage, role-wise data augmentation process for KD. This process consists of: (1) training a teacher network till convergence while learning a schedule of policies to augment the training data specifically for the teacher; (2) distilling the knowledge from the teacher into a student network while learning another schedule of policies to augment the training data specifically for the student. It is worth noting that this two-stage framework is orthogonal to existing methods for KD, which focus on how the knowledge to be distilled is defined; thus, our approach can be combined with previous methods straighforwardly.

Although our proposed method can in principle be applied to any models trained via KD, we focus specifically on how to use it to transfer the knowledge from a full-precision teacher network into a student network with lower bit-width. Network quantization is crucial when deploying trained models on embedded devices, or in data centers to reduce energy consumption (Strubell et al., 2019). KD-based quantization (Zhuang et al., 2018; Polino et al., 2018) jointly trains a full-precision model, which acts as the teacher, alongside a low-precision model, which acts as the student. Previous work has shown that distilling a full-precision teacher's knowledge into a low-precision student, followed by fine-tuning, incurs noticeable performance degradation, especially when the bit-widths are below four (Zhuang et al., 2018; Polino et al., 2018). We show that it is advantageous to use adaptive data augmentation to generate more training data for the low-precision network based on its specific weaknesses. For example, low-precision networks may have difficulties learning rotation-related patterns,[1] and the data augmentation agent should be aware of this and generate more such data points. One positive side-effect for demonstrating the effectiveness of our method is that the improvement brought by our proposed method is more significant compared to the experiments on all full-precision models.

## 2 RELATED WORK

**Knowledge distillation.** KD is initially proposed for model compression, where a powerful wide/deep teacher distills knowledge to a narrow/shallow student to improve its performance (Hinton et al., 2014; Romero et al., 2015). In terms of the definition of knowledge to be distilled from the teacher, existing models typically use teacher's class probabilities (Hinton et al., 2014) and/or intermediate features (Romero et al., 2015; Park et al., 2019; Seunghyun Lee, 2019; Hyun Lee et al., 2018). Among those KD methods that utilize intermediate feature maps, Relational KD (RKD) considers (Park et al., 2019) the intra-relationship in the same feature map, while Multi-Head KD (MHKD)(Seunghyun Lee, 2019) and KD using SVD (KD-SVD) (Hyun Lee et al., 2018) utilize the inter-relationship across feature maps. By contrast, we propose to incorporate both the intra- and inter-relationships within and across feature maps.

**Automated data augmentation.** Manually applying data augmentation rules such as random rotating, flipping, and scaling are common practices for training neural models on image classification tasks (Krizhevsky et al., 2012; He et al., 2016). Several recent works attempt to automate the data augmentation process. Generative adversarial networks (Ratner et al., 2017) and Bayesian optimization (Tran et al., 2017) have been used for this process. DeVries & Taylor (2017) augment training data in the learned feature space by injecting noise and interpolation. Lemley et al. (2017) learn how to combine pairs of images for data augmentation. AutoAugment (Cubuk et al., 2018) searches for the optimal data augmentation policies (e.g., how to rotate) based on reinforcement learning. However, the search process is computationally expensive. Population-based augmentation (PBA) (Ho et al., 2019) uses an evolution-based algorithm to automatically augmenting data in an efficient way. In contrast to previous approaches, we study the effect of the training data for KD and propose to use automatic data augmentation to train the student better from her teacher.

---

[1]We will visualize the learned schedules of policies in Section 5.5.

## 3 PRELIMINARIES

### 3.1 POPULATION-BASED AUGMENTATION (PBA)

PBA (Ho et al., 2019), as an evolutionary search algorithm, learns a dynamic *per-epoch* schedule of augmentation policies, denoted as $\mathcal{A}$. Since this schedule is epoch-based, it will *re-create* the augmented dataset every epoch. More concretely, PBA begins with a population of models that are trained in parallel on a small subset of the original training data. The weights of the worst performing models in the population are replaced by those from better performing models (i.e., exploitation), and the policies are mutated to new ones within the pre-defined policy search space (i.e., exploration). After training, PBA usually keeps the learned augmentation schedule of policies but *discards* the elementary parameters of the models. A different model (e.g., a larger one) can then use the learned schedule to improve its training on the same task.

### 3.2 KNOWLEDGE DISTILLATION (KD)

Following the notations in (Park et al., 2019), a KD method aims to minimize the objective function

$$\mathcal{L}_{\text{general}} = \mathcal{L}_{\text{task}} + \lambda \cdot \mathcal{L}_{\text{KD}}, \tag{1}$$

where $\lambda$ is a hyper-parameter to balance the impact of the KD loss term.

In this paper, for classification tasks, $\mathcal{L}_{\text{task}} = \sum_{x_i \in \mathcal{X}} \mathcal{H}(\text{softmax}(\mathcal{F}_S^{\text{final}}(x_i)), y_{\text{truth}})$, where $\mathcal{X}$ refers to training sample space, $y_{\text{truth}} \in \mathcal{Y}$ are the ground-truth labels, $\mathcal{F}_S(\cdot)$ is the student network, and $\mathcal{H}(\cdot)$ denotes the cross-entropy.

The KD term can be defined as

$$\mathcal{L}_{\text{KD}} = \sum_{x_i \in \mathcal{X}} l(\mathcal{F}_T(x_i), \mathcal{F}_S(x_i)), \tag{2}$$

where $\mathcal{F}(\cdot)$ is the function of the network and $l(\cdot)$ is a loss function to compute the difference between the teacher network and the student network.

For KD methods that use soft labels (Hinton et al., 2014), the objective can be defined as

$$\mathcal{L}_{\text{KD}}^{\text{soft}} = \sum_{x_i \in \mathcal{X}} \mathcal{H}(\text{softmax}(\mathcal{F}_T^{\text{final}}(x_i)), \text{softmax}(\mathcal{F}_S^{\text{final}}(x_i))), \tag{3}$$

where $\mathcal{F}^{\text{final}}(x_i)$ is the feature map of the final layer.

There exist some KD methods that utilize the intermediate feature maps in complementary ways. For example, Relational KD (Park et al., 2019) considers the intra-relationships. That is, given the feature map of layer $j$, the KD loss can be formulated as:

$$\mathcal{L}_{\text{KD}}^{\text{intra}} = \sum_{x_i \in \mathcal{X}} l(\Phi(\mathcal{F}_T^j(x_i)), \Phi(\mathcal{F}_S^j(x_i))), \tag{4}$$

where $\Phi(\cdot)$ refers to the potential function measuring the pairwise relationship inside a feature map from student network or teacher network and $\mathcal{F}^j(x_i)$ is the feature map of layer $j$ (which may include the final logits layer). Therefore, this feature-based KD method includes the benefits of using soft labels.

On the other hand, some works (Seunghyun Lee, 2019; Hyun Lee et al., 2018) consider the inter-relationships, where the KD term can be formulated as:

$$\mathcal{L}_{\text{KD}}^{\text{inter}} = \sum_{x_i \in \mathcal{X}} l(\varphi(\mathcal{F}_T^j(x_i), \mathcal{F}_T^k(x_i)), \varphi(\mathcal{F}_S^j(x_i), \mathcal{F}_S^k(x_i))). \tag{5}$$

Here, $\varphi(\cdot)$ measures the inter-relationship between feature maps of different layers, i.e. $k \neq j$.

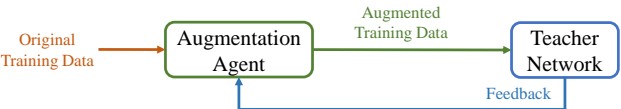

Figure 1: Concept diagram (stage-$\alpha$) of training a data augmentation agent for the $\mathcal{N}_T$.

### 3.3 QUANTIZATION

In this work, we use DoReFa[2] (Zhou et al., 2016) to quantize both weights and activations. The quantization function $Q(\cdot)$ is defined as:

$$r_q = Q(r) = \frac{1}{2^{n_{\text{bits}}} - 1} \cdot \text{round}((2^{n_{\text{bits}}} - 1) \cdot r), \tag{6}$$

where $r$ is the full-precision value, $r_q$ indicates the quantized value, $n_{\text{bits}}$ refers to the number of bits to represent this value. With this quantization function, the quantization on weights $w$ is defined as:

$$w_q = 2 \cdot Q(\frac{\tanh(r)}{2 \cdot \max(|\tanh(w)|)} + \frac{1}{2}) - 1. \tag{7}$$

The back-propagation is approximated by the straight-through estimator (Bengio et al., 2013) and the partial gradient $\frac{\partial l}{\partial r}$ w.r.t. the loss $l$ is computed as:

$$\frac{\partial l}{\partial r} = \frac{\partial l}{\partial r_q} \cdot \frac{\partial r_q}{\partial r} \approx \frac{\partial l}{\partial r_q}. \tag{8}$$

## 4 THE PROPOSED METHOD

Our proposed method has two stages, which will be described in the following subsections. In the first stage, we train a teacher network, denoted as $\mathcal{N}_T$, with the help of PBA-based augmentation. In the second stage, we further distill the knowledge from $\mathcal{N}_T$ (pre-trained in the first stage) to the student network, denoted as $\mathcal{N}_S$, while learning another augmentation schedule to augment the training data for $\mathcal{N}_S$.

### 4.1 STAGE-$\alpha$

In general, a teacher can provide better training signals for the student if the teacher's performance increases (Mirzadeh et al., 2019). As shown in Fig. 1, we apply PBA to learn a dynamic per-epoch schedule of augmentation policies, $\mathcal{A}_T$, for $\mathcal{N}_T$ on a small subset of training data. That is, the augmentation agent's training signal is defined as the feedback of $\mathcal{N}_T$'s accuracy on a subset of the dataset. After this, we use the discovered schedule $\mathcal{A}_T$ to augment the whole training dataset and re-train $\mathcal{N}_T$ on it till convergence.

### 4.2 STAGE-$\beta$

KD methods have shown to be effective at improving the performance of lower-capacity networks using the knowledge from higher-capacity networks. In order to take advantage of this functionality, we apply the KD methods together with data augmentation in stage-$\beta$, as shown in Fig. 2.

More concretely, we first use PBA to learn an epoch-based augmentation schedule $\mathcal{A}_S$ for $\mathcal{N}_S$ on a subset of the dataset. Different from the schedule $\mathcal{A}_T$ learned in stage-$\alpha$, $\mathcal{A}_S$ is learned based on the feedback (i.e., accuracy) from $\mathcal{N}_S$, which is trained with KD. In other words, $\mathcal{N}_S$ receives additional training signals from $\mathcal{N}_T$ that is pre-trained in stage-$\alpha$. We then use the learned $\mathcal{A}_S$ to augment the whole training dataset, and re-train $\mathcal{N}_S$ on it with the distilled knowledge from $\mathcal{N}_T$. Note that, because the learned schedule is epoch-based, we do not use the discovered schedule $\mathcal{A}_T$ from stage-$\alpha$ to augment the training data as initialization.

---

[2]It should be noted that our proposed method is orthogonal to any particular quantization method.

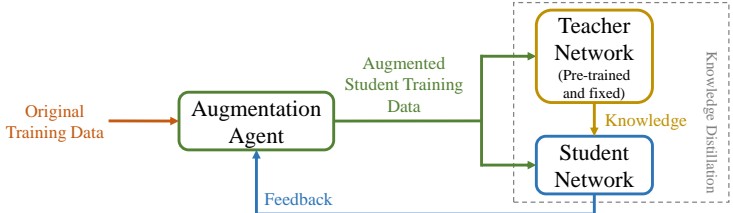

Figure 2: Concept diagram (stage-$\beta$) to augment training datapoints for both $\mathcal{N}_T$ and $\mathcal{N}_S$. The $\mathcal{N}_T$ has been pre-trained using the method shown in Section 4.1, and is fixed during training. The augmentation agent in stage-$\beta$ is designed to learn schedules of polices that are different from those learned in stage-$\alpha$, and thus the agent only receives the feedback from $\mathcal{N}_S$.

When $\mathcal{N}_S$ is a low-precision network, following (Furlanello et al., 2018), we share the same network architecture[3] between $\mathcal{N}_T$ and $\mathcal{N}_S$. When $\mathcal{N}_S$ is a full-precision network, it will have fewer layers compared to $\mathcal{N}_T$.

## 5 EXPERIMENTS

### 5.1 SETTINGS

We evaluate our approach on two benchmark datasets: CIFAR-10 (Krizhevsky et al., 2009) and CIFAR-100. CIFAR-10 consists of 60,000 32×32 color images in 10 classes, with 6000 images per class and CIFAR-100 has 100 classes with 600 images per class. Both have 50,000 training images and 10,000 test images.

We search over a "reduced" CIFAR-10/CIFAR-100 with 4,000 training images and 36,000 validation images, which is the same as in (Ho et al., 2019). All the data augmentation models are run with 16 total trials to generate augmentation schedules. Following PBA, in stage-$\alpha$, we run PBA to create schedules over separate models and then transfer the CIFAR-10 policy to CIFAR-100. However, for student network training in stage-$\beta$, we empirically use the respective "reduced" dataset. The data augmentation approaches for the baselines include random crop and horizontal flipping operations. Following (Ho et al., 2019), our policy search space has a total of 15 operations, each having two magnitude and discrete probability values. We use discrete probability values from 0% to 100%, in increments of 10%. Magnitudes range from 0 to 9.

The models we evaluate on include AlexNet (Krizhevsky et al., 2012) and ResNet18 (He et al., 2016). The number of epochs is 200 and the batch size is set to 128. For full-precision network $\mathcal{N}_T$, the learning rate starts from 0.1 and is decayed by 0.1 after every 30% of the total epochs. we use SGD with a Nesterov momentum optimizer. The weight decay is set to $5 \times 10^{-4}$. For quantization, the learning rate is set to $10^{-3}$ and is divided by 10 every 30% of the total epochs. We use the pre-trained teacher network model as the initial point of student network. We use a smaller weight decay $10^{-5}$ assuming that less regularization is needed by the lower-precision networks. Following DoReFa (Zhou et al., 2016), the first layer and last layer are not quantized.

Following Clark et al. (2019), during training, we gradually transit the student from learning based on the teacher to training based on the ground-truth labels. This heuristic provides the student with more rich training signals in the early stage but does not force the student to strictly mimic the teacher's behaviors. As for the implementation, we decay the balancing hyper-parameter $\lambda$ in the KD loss by 0.5 every 60 epochs.

### 5.2 COMPARING DIFFERENT KD METHODS

As mentioned in Section 3.2, there exist complementary KD methods considering both intra- and inter-relationships within and across feature maps. A natural question is if it would be beneficial

---

[3]Note this is not a hard constraint, we choose such strategy to reduce the number of factors that might influence the final performance.

to combine them to further boost the performance together with data augmentation. Therefore, we propose a simple extension to these complementary KD methods, dubbed as II-KD, by incorporating intra-relationships inside the feature map and inter-relationships across different feature maps. We incorporate the two relationships into the final objective function as follows:

$$\mathcal{L}_{\text{KD}}^{\text{II}} = \mathcal{L}_{\text{original}} + \lambda \cdot (\mathcal{L}_{\text{KD}}^{\text{intra}} + \mathcal{L}_{\text{KD}}^{\text{inter}}), \qquad (9)$$

where we only use a single balancing hyper-parameter $\lambda$ between the original loss and the distillation loss, which does not introduce extra hyper-parameters.

More precisely, our KD method incorporates components of three conventional KD methods: RKD (Park et al., 2019), MHGD (Seunghyun Lee, 2019) and KD-SVD (Hyun Lee et al., 2018). As shown in Eq. (9), we add the three KD terms together with equal coefficients. We use the loss function $l(\cdot)$ following their approaches. For the back-propagation, we clip the gradient for KD loss as in KD-SVD, because this will smoothly post-processes the gradient to limit the impact of KD loss in training. For AlexNet we select the feature maps of ReLU layers after the convolution/max pooling layer. For ResNet18, we select the feature maps of the last ReLU layer of each residual block.

We evaluate our proposed KD extension on CIFAR-100 with ResNet18 for different bit-width settings by comparing with various KD methods. For the baseline methods, we use their default settings with a fixed and pre-trained teacher network in the training stage and $\lambda = 1$ for the knowledge distillation loss. We set $\lambda = 0.4$ for II-KD in Eq. (9), as we have two KD terms. Tab. 1 reports the results on various augmented KD methods. We observe that our proposed methods clearly outperforms the other KD methods on all the settings, though the improvements over MHGD and KD-SVD are not huge. The results also reveal that only relying soft labels is not as effective as utilizing multiple supervising signals from the teacher.

Table 1: Accuracy (%) on CIFAR-100 with ResNet18 using different KD methods. We compare ours with the following methods: Soft labels (Hinton et al., 2014), DML (Zhang et al., 2018), RKD (Park et al., 2019), MHGD (Seunghyun Lee, 2019) and KD-SVD (Hyun Lee et al., 2018).

| Bit-Width (Weight / Activation) | Soft labels | DML | RKD | MHGD | KD-SVD | II-KD |
|---|---|---|---|---|---|---|
| 4/4 | 70.48 | 72.47 | 71.84 | 73.52 | 73.92 | 74.21 |
| 2/2 | 70.09 | 69.72 | 70.71 | 71.80 | 72.97 | 73.35 |

### 5.3 Is Role-Wise Augmentation with KD Effective for Quantization?

In this subsection, we aim to answer this question: is our two-stage role-wise augmentation with KD effective for network quantization? We conduct experiments on CIFAR-10 and CIFAR-100 datasets under full-precision, 4-bit, and 2-bit settings.

From Tab. 2, we can observe that training with learned data augmentation schedules does not improve the performance of low-precision networks too much. Similar to the results obtained in (Zhuang et al., 2018), transferring knowledge from the full-precision to the low-precision student usually helps the training of students, which is especially obvious on the CIFAR-100 dataset. Tab. 2 also clearly shows that our proposed pipeline consistently improves the performance of the low-precision student networks. For example, the 4-bit $\mathcal{N}_S$ is comparable with full-precision reference without loss of accuracy for CIFAR-10 and with loss of accuracy within 1.0% on CIFAR-100. When decreasing the precision to 2-bit, the results are still promising as compared with other baselines, even though there is a performance gap between the 2-bit and the full-precision models. For instance, our approach usually outperforms the strong baseline, only using II-KD, by more than 1.0%.

### 5.4 Comparing Schedules

Here we aim to answer this question: how effective is it if we use $\mathcal{A}_T$, learned based on the feedback from $\mathcal{N}_T$ in stage-$\alpha$, to dynamically augment the training dataset and train $\mathcal{N}_S$ on it? Tab. 3 reports the accuracy comparison with different KD methods and augmentation schedules. We can clearly see that augmenting the training dataset for $\mathcal{N}_S$ with $\mathcal{A}_S$ consistently outperforms those using the transferred schedules $\mathcal{A}_T$ among different KD methods. This observation is consistent with our assumption that $\mathcal{N}_S$ has her own optimal augmentation schedule, $\mathcal{A}_S$, that is different from $\mathcal{A}_T$ for $\mathcal{N}_T$. In particular, blindly applying the teacher augmentation schedule $\mathcal{A}_T$ may negatively influence

Table 2: Accuracy (%) on CIFAR-10 and CIFAR-100 datasets with different bit-widths. **Vanilla Training** for 4-bit and 2-bit refers to training a network based on DoReFa (Zhou et al., 2016) from scratch without learned data augmentation. **Teacher after Stage-$\alpha$** refers to using learned schedules discovered by PBA to *re-train* $\mathcal{N}_T$ as described in Section 4.1. **Student with only II-KD** refers to training $\mathcal{N}_S$ using II-KD but without the learned data augmentation. **Student after Stage-$\beta$** refers to training $\mathcal{N}_S$ using II-KD and the learned data augmentation. For **Vanilla Training** and **Teacher after Stage-$\alpha$**, we report the accuracy of $\mathcal{N}_T$, and for the rest we report the accuracy of $\mathcal{N}_S$.

| Methods | | AlexNet CIFAR-10 | AlexNet CIFAR-100 | ResNet18 CIFAR-10 | ResNet18 CIFAR-100 |
|---|---|---|---|---|---|
| | 32-bit | 90.58 | 65.80 | 93.57 | 74.85 |
| Vanilla Training | 4-bit | 89.72 | 60.25 | 90.97 | 69.81 |
| | 2-bit | 88.77 | 58.96 | 90.00 | 67.06 |
| | 32-bit | 91.62 | 66.40 | 94.49 | 75.19 |
| Teacher after Stage-$\alpha$ | 4-bit | 90.06 | 60.65 | 91.47 | 70.24 |
| | 2-bit | 89.28 | 58.59 | 89.99 | 67.32 |
| Student with only II-KD | 4-bit | 90.55 | 65.55 | 91.42 | 73.85 |
| | 2-bit | 89.18 | 63.49 | 90.60 | 72.44 |
| Student after Stage-$\beta$ | 4-bit | 92.00 | 65.69 | 94.44 | 74.21 |
| | 2-bit | 90.63 | 64.06 | 93.20 | 73.35 |

the training of $\mathcal{N}_S$ as compared to only using KD. For example, the learned schedule based on the teacher $\mathcal{A}_T$ degrades the performance of $\mathcal{N}_S$ by 0.58% for AlexNet on CIFAR-100 as compared to applying KD methods, as shown in Tab. 2.

Table 3: Accuracy (%) on CIFAR-100 with 4-bit networks using different KD methods.

| Methods | AlexNet | | | ResNet18 | | |
|---|---|---|---|---|---|---|
| | DML | MHGD | Ours | DML | MHGD | II-KD |
| Schedule based on teacher | 61.61 | 61.62 | 64.97 | 71.78 | 69.76 | 73.46 |
| Schedule based on student | 63.73 | 63.47 | 65.69 | 72.47 | 73.52 | 74.21 |

## 5.5 ANALYZING THE LEARNED SCHEDULES

To analyze the difference on the discovered schedules between $\mathcal{N}_T$ (*i.e.,* full-precision ResNet18) and $\mathcal{N}_S$ (*i.e.,* 4-bit ResNet18), we report their augmented schedules quantitatively in terms of normalized probability and magnitude on CIFAR-100 in Fig. 3. We normalize the probability of each epoch by dividing the maximal summation of probabilities for all operations across all epochs.

It can be seen that the discovered schedules $\mathcal{A}_S$ for $\mathcal{N}_S$ is quite different from $\mathcal{A}_T$ for $\mathcal{N}_T$. In particular, for $\mathcal{A}_T$, there is an emphasis on Brightness, Posterize, Rotate, Sharpness and TranslateY, while $\mathcal{A}_S$ cares more about Contrast, ShearX and TranslateY. Furthermore, we observe that the probability and magnitude increase as the epoch evolves. For $\mathcal{A}_S$, in the beginning, KD plays a more important role, and there is no augmentation operation before about epoch 50. As the training continues, the augmentation policies become more important. One possible reason is that, for low-precision networks, KD methods can provide rich training signals such that data augmentation does not help in the early training phases.

Furthermore, we observe that, compared to $\mathcal{A}_T$, the schedule for student $\mathcal{A}_S$ evolves more smoothly in the sense that the policy updating frequency is lower. For example, the probability and magnitude values change about every 40 epochs for student, while the policies for teacher update about every 15 epochs. One possible reason is that for the low-precision $\mathcal{N}_S$, KD methods make the training process more smooth and it is not necessary to change the augmentation policies too frequently. This is consistent with the observations shown in Tab. 2 that KD can already provide useful training signals. Also, this validates our assumption that $\mathcal{N}_S$ has her own optimal augmentation schedule $\mathcal{A}_S$ that is different from $\mathcal{A}_T$.

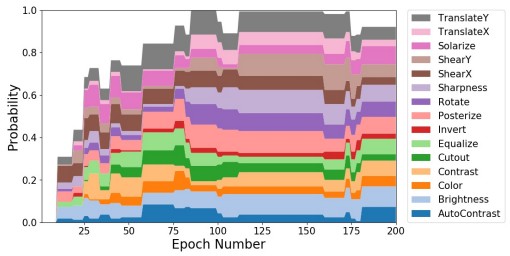

(a) Normalized plot of operation probability parameters over time for the teacher network $\mathcal{N}_T$.

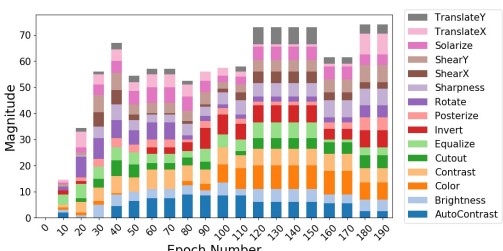

(b) Operation magnitude parameters over time for the teacher network $\mathcal{N}_T$.

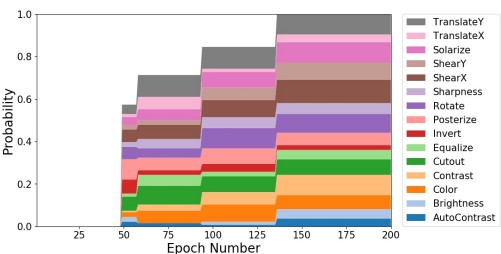

(c) Normalized plot of operation probability parameters over time for the student network $\mathcal{N}_S$.

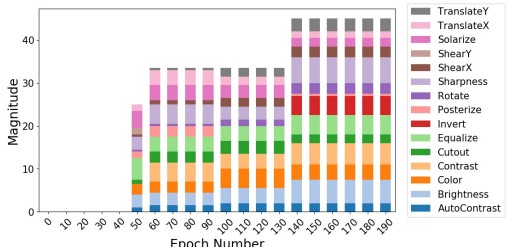

(d) Operation magnitude parameters over time for the student network $\mathcal{N}_S$.

Figure 3: Evolution of magnitude and probability parameters in the learned schedules. Each operation appears in the parameter list twice, and we take the mean values of the parameter.

## 5.6 FURTHER COMPARISON ON FULL-PRECISION NETWORKS

This subsection aims to verify the effectiveness of our proposed methods on more conventional settings where both $\mathcal{N}_T$ and $\mathcal{N}_S$ are full-precision networks. We select ResNet18 as the teacher, and ResNet8 as the student to check how our proposed methods affect the student network.

Tab. 4 shows that our proposed method outperforms the standard baseline training. The discovered augmentation schedule further boosts the performance of shallow $\mathcal{N}_S$ based on II-KD, though the improvement is not that significant compared with the results obtained when $\mathcal{N}_S$ is low-precision network. This shows that our proposed method can be used for full-precision training tasks.

Table 4: Accuracy (%) on CIFAR-100 with full-precision ResNet8 as $\mathcal{N}_S$ and full-precision ResNet18 as $\mathcal{N}_T$ under different settings. **Vanilla Training** refers to training a full-precision network from scratch. **Re-Training with PBA** refers to using learned schedules discovered by PBA to *re-train* $\mathcal{N}_T$ as described in Section 4.1. **Student with only II-KD** refers to training $\mathcal{N}_S$ using II-KD but without the learned data augmentation. **Student after Stage-$\beta$** refers to training $\mathcal{N}_S$ using II-KD and the learned data augmentation.

|  | Vanilla Training | Re-Training with PBA | Student with only II-KD | After Stage-$\beta$ |
|---|---|---|---|---|
| Accuracy | 74.31 | 74.52 | 75.52 | 75.88 |

## 6 CONCLUSION

Previous literature on KD focuses on exploring the knowledge representation and the strategies for distillation. However, both the teacher and student learn from the same training data without adapting the different learning capabilities. To address this issue, we propose customizing distinct agents to automatically augment the training data for the teacher and student, respectively. We have extensively studied the effect of combining data augmentation and knowledge distillation. Furthermore, we propose a simple feature-based KD variant that incorporates both intra- and inter-relationships within and across feature maps. We have empirically observed that the student can learn better from the teacher with the proposed approach, especially in the challenging low-precision scenario, and the learned schedules are different for the teacher and student.

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
