# OpenReview forum: "Role-Wise Data Augmentation for Knowledge Distillation"
_ICLR.cc/2020/Conference — Reject_

### Official Review · AnonReviewer3 · 2019-10-19
**Official Blind Review #3**

**Rating:** 3

**Review:**

This paper takes the idea of Population-Based Augmentation (PBA) and extends it to knowledge distillation (KD). The idea is that the ideal augmentation protocol for training-from-scratch (or in this context teacher training) may not be the best for student networks under a KD loss.

I am borderline about this paper and had to pick one, so I landed on Weak Reject. On one hand, I think it’s a really neat idea to apply PBA in this context, and package it as a strage-alpha/stage-beta training procedure. However, the experimental results seem very incremental and I’m not convinced there is a genuine signal there.

Experiments:

Table 1 offers a great comparison of prior work, as well as the combination of prior work (II-KD). Table 2 tells me that PBA seems incremental both for the teacher and the student. Going from vanilla training to student with II-KD gets you most of the way there, and the primary contribution of this paper just gives you a slight benefit above this. It’s great that a more traditional full-precision comparison was also added in Table 4, but this table also confuses me. First, it was unclear if “vanilla training” referred to the teacher or the student. The number 74.31 is not the same as in Table 2 (74.85), so I assume this means it’s the student? If so, the student is already very close to the teacher, and this is not a great starting point for evaluating KD. The student after stage-beta also outperforms the teacher - something that was mentioned in the related work, but I would like more discussion around it specifically for Table 4. Another thing I was wondering was how important PBA is for the teacher’s ability to be a good teacher. It gives a modest boost in Table 2; what if we skip PBA in the teacher but still do it for the student. This would be interesting to add.

Overall, there aren’t that many experiments. The dataset is never more challenging than CIFAR-100. There are also no error bars, which are particularly important when the improvements are small. As for Figure 3, I don’t know if there is anything intuitive we can glean from this. I may just be variation between experiments, as far as I can tell.

I think this paper can be made stronger by making the experimental evidence broader, as well as the analysis of why this works stronger. Without these improvements, the reader is left wondering if there really is any significant benefit. We have to remember that PBA is not cheap (perhaps much cheaper than AutoAugment, but more expensive than fixed augmentation). For most practitioners, the complication and compute costs of PBA would probably not be worth adding on top of KD, if the benefits are too modest.

Minor:

In Table 3, it says "Ours" for AlexNet and "II-KD" for ResNet8. Should the both be the same?

**Experience Assessment:**

I have published one or two papers in this area.

**Review Assessment: Checking Correctness Of Derivations And Theory:**

I assessed the sensibility of the derivations and theory.

**Review Assessment: Checking Correctness Of Experiments:**

I assessed the sensibility of the experiments.

**Review Assessment: Thoroughness In Paper Reading:**

I read the paper at least twice and used my best judgement in assessing the paper.

---

> ### Author Response · Authors · 2019-11-13
> **Responses to Reviewer #3**
>
> Thank you very much for your helpful review. Please refer to our general responses thread for some common concerns raised by multiple reviewers. In this thread, we address your specific questions.
>
> Regarding your concerns, here are our responses:
>
> We are running more models on more challenging datasets (e.g. ImageNet). We will also provide more visualization of the learned policies for students on more problems to ensure that the current behavior shown in Figure 3 is not just a variation.

---

### Official Review · AnonReviewer2 · 2019-10-23
**Official Blind Review #2**

**Rating:** 3

**Review:**

The authors propose a new method to distill a teacher model into a student model. They demonstrate improvements over existing distillation variants. The results are impressive for low-precision networks.

However, I found a few problems with the paper:

In the author’s methods, there seem to be multiple steps:
1. Train the teacher with PBA (stage-alpha).
2. Train the student with PBA, using a subset of the data and then using the teacher to learn the augmentation policy (stage-beta). This is described in Section 4.2, but I found it incredibly confusing to get a clear picture since there are several moving parts here (augmentation, student solo training, distillation with teacher).
3. Additionally in Section 5.2 (experiments), the authors propose the combined inter/intra distillation loss, which is named ‘II-KD’.

In Table 3, the accuracy of ResNet18 on CIFAR-100 with II-KD is the same as accuracy of Resnet18 Student after stage-beta in Table 2. Same for AlexNet on CIFAR-100. Which of the tables are wrong?

Also, in section 5.1, the authors mention that they use the pre-trained teacher as a starting point for the student network. This is not a fair comparison with the ‘Soft Labels’ approach of Hinton et al, where the student network is not initialized from the teacher.

Further, it is unclear why the augmentation plot in the student in Figure 3(c) differs wildly from the augmentation plot of the teacher. There is no augmentation for the first 50 epochs, and then the probability of all the augmentation operations moves in discrete steps together.

The novelty in the paper seems to be:
a) Applying PBA to a Distillation setting.
b) Introducing the ‘II-KD’ loss.

As mentioned (a) is not explained clearly. (b) is explained in a generic way but the authors do not give an example of the inter/intra feature map comparisons.

Overall, while I feel the results are impressive, the method is complex as it is presented. I would be reluctant to accept the paper without further clarification from the authors.

**Experience Assessment:**

I have read many papers in this area.

**Review Assessment: Checking Correctness Of Derivations And Theory:**

I assessed the sensibility of the derivations and theory.

**Review Assessment: Checking Correctness Of Experiments:**

I assessed the sensibility of the experiments.

**Review Assessment: Thoroughness In Paper Reading:**

I read the paper at least twice and used my best judgement in assessing the paper.

---

> ### Author Response · Authors · 2019-11-13
> **Responses to Reviewer #2**
>
> Thank you very much for your helpful review. Please refer to our general responses thread for some common concerns raised by multiple reviewers. In this thread we address your specific questions.
>
> Regarding your concerns, here are our responses:
>
>
> Q1: Multiple steps
> A1: The backbone of the framework is a conventional KD process: train a teacher and then distill the knowledge of the teacher to a student. The key extra component is to learn to augment the training data of the student and teacher differently.
>
> More concretely:
> 1. We learn how to augment the teacher and train the teacher until convergence.
> 2. We learn how to augment the data (from scratch) while distilling the teacher’s knowledge into the student.
>
> Introducing the II-KD is to make sure that the baseline model is as strong as possible, and it is not essential.
>
> Q2: Results in Table 2 and Table 3
> A2: There is no error, and the results in Table 3 are the same as some of them in Table 2. The purpose of Table 3 is to do some comparison in another dimension. More specifically, In Table 3, we report the performance of Stage-with different KD methods but not pure KD methods.
>
> Q3: Pre-training teacher
> A3: For all the comparisons, we always pre-train the teacher. We did not just copy the results obtained from other papers. Instead, we re-implemented all the methods and run them ourselves and report here.
>
> Q4: There is no augmentation for the first 50 epochs
> A4: We have a dedicated subsection (5.5) discussing this behavior. One possible reason is that for the low-precision student, KD methods make the training process more smooth and it is not necessary to change the augmentation policies too frequently.
> Actually, in the original PBA paper, the learned augmentation policy does not perform any augmentation operations in the early stage (before epoch 12).
>
>
> Q5: Comparison of inter/intra feature maps
> A5: We will add this in the next version soon.

---

### Official Review · AnonReviewer4 · 2019-11-01
**Official Blind Review #4**

**Rating:** 6

**Review:**

The authors hypothesise that, in a Knowledge Distillation (KD) setting, the student network may benefit from learning from different training data than the teacher. The motivation comes from the fact that human teachers "adapt" the examples given their students, depending on their individual expertise on the subject, personal and cultural biases, and other factors.

More concretely, the paper proposes to use an evolutionary algorithm for data augmentation (PBA, already published) to train the teacher and student networks. The key aspect of the proposed method is that the teacher and the student use different augmentation schedules. The augmentation schedules improve results on both the teacher and student networks.
Orthogonally to this, they also propose to combine different KD objective functions and show that this also improves the results over each loss in isolation.

The datasets used in the experiments are CIFAR-10 and CIFAR-100. In most of the experiments the student has a (much) lower bit-width than the teacher, but they also apply the method with a student network with less parameters. Two different network architectures are used in the experiments: AlexNet and Resnet18, their method shows consistent improvements over the baselines in all cases (although some improvements may be considered marginal).

Overall the paper is well motivated, clearly written and, in the experiments section, they give enough details, and/or cite previous works that contain them, to reproduce the experiments.

Most importantly, the experiments are well designed to test the initial hypothesis:
1. The authors show that the student (and teacher) trained with the PBA data augmentation achieves a higher accuracy than the baseline method (Table 2). However, this is not enough to confirm/refute the hypothesis itself, since it is known that data augmentation generally helps.
2. They show that the two augmentation strategies are different, by using the teacher’s augmentation to train the student network (Table 3). The results show that it’s better to use a specific data augmentation schedule for the student network.

However, (the main criticism is that) the paper only partially confirms the hypothesis. For instance, it is not clear whether the hypothesis is true for other forms of KD, or it only applies when training a student with lower bit-width. In Section 5.6, they train a student with fewer parameters (less layers) than the teacher, instead of fewer bits/parameter. However, the improvements of their method over the baseline (II-KD) seem marginal there. And most importantly, as pointed out earlier, this alone is not enough to confirm the hypothesis. A similar table to Table 3 should be included in this section.

Secondly, the proposed KD loss, which is just a combination of previously published works, is an orthogonal improvement to the main method (as the authors admit), and it is not related at all to the subject of the study. It is obviously good to introduce more than one contribution in a paper, but this second (and minor) contribution should be well motivated in its own, in order to avoid "distracting" the reader from the main contribution.

In addition, I would suggest to perform statistical tests to give additional robustness to the conclusions drawn from the experiments. They perform experiments on different architectures (AlexNet and Resnet18) and different KD losses, and the conclusions are always consistent with the hypothesis, but it’s not clear whether the improvements are statistically significant or not. In this regard, it would also be appreciated to include results with other datasets, since only CIFAR-10 and CIFAR-100 (which are very similar datasets) were used.

Beyond statistical significance, it’s also not clear how important are these results for researchers working outside the scope of Knowledge Distillation.

Finally, some minor comments:

- L_original in Eq. (9) is not defined. I'm assuming it's the L_{KD}^{soft} loss.
- Please, check consistency of pronouns: authors refer to the teacher network as “it” (e.g. “[...] teacher distills knowledge to a narrow/shallow student to improve its performance”, page 2), but to the student as “she” (e.g. “[...] to train the student better from her teacher”, page 2).
- “our methods clearly outperforms [...]” (page 6)  -> “our method clearly outperforms [...]”.

Score: Borderline accept, but I will increase the score if the authors address my concerns and provide better evidence that the hypothesis is confirmed.

**Experience Assessment:**

I do not know much about this area.

**Review Assessment: Checking Correctness Of Derivations And Theory:**

N/A

**Review Assessment: Checking Correctness Of Experiments:**

I assessed the sensibility of the experiments.

**Review Assessment: Thoroughness In Paper Reading:**

I read the paper thoroughly.

---

> ### Author Response · Authors · 2019-11-13
> **Responses to Reviewer #4**
>
> Thank you very much for your helpful review. Please refer to our general responses thread for some common concerns raised by multiple reviewers. In this thread we address your specific questions.
>
> Regarding your concerns, here are our responses:
>
> 1. It should be noted that II-KD is already a very strong baseline, which outperforms many others. It is true that the improvements in the case of having fewer parameters are not that significant, and we are working on providing further experiments on Imagenet dataset.
> 2. In the original submission, we have stated and will state here again, that the introduction of II-KD is never the main contribution and stems from our efforts to make our baseline KD method as strong as possible.
> 3. We will perform multiple experiments to provide at least mean and variance for each setting.

---

### Author Response · Authors · 2019-11-13
**General Responses**

1. We are in the process of conducting more experiments including running experiments on a larger scale and more complex datasets such as ImageNet.
2. We provide II-KD as part of our implementation details, rather than the main contribution. We believe it is fair to use it so that the baseline systems we compare with is stronger.
3. We will improve our analysis section by providing a clearer explanation.

---

### Decision · Program_Chairs · 2019-12-19

**Decision:**

Reject

**Comment:**

This paper studies Population-Based Augmentation in the context of knowledge distillation (KD) and proposes a role-wise data augmentation schemes for improved KD. While the reviewers believe that there is some merit in the proposed approach, its incremental nature and inherent complexity require a cleaner exposition and a stronger empirical evaluation on additional data sets. I will hence recommend the rejection of this manuscript in the current state. Nevertheless, applying PBA to KD seems to be an interesting direction and we encourage the authors to add the missing experiments and to carefully incorporate the reviewer feedback to improve the manuscript.